# UNMASKING THE TINY: FOREGROUND PROBING FOR SMALL OBJECT DETECTION

## ABSTRACT

Detecting small objects in high-resolution images is challenging, as small targets are often overwhelmed by the surrounding background and thus prone to being missed or misclassified. To address this issue, this work proposes a *foreground probing* paradigm to recover and refine suppressed foreground scores by leveraging collective classification features. At its core, we introduce a *sparse token selection module* (STSM) that identifies potential foreground tokens across feature maps, for the sake of preserving promising candidates from being submerged within background features. To further enhance these representations, we design a *foreground refinement module* (FRM) that distills a semantically enriched attention map from classification features to guide information aggregation. This allows tokens to adaptively reference semantically similar neighbors, thereby strengthening discrimination between foreground and background in complex scenes. Extensive experiments demonstrate that our method achieves a superior performance on small object detccion. Our code will be released.

## 1 INTRODUCTION

High-resolution small object detection is increasingly vital for critical applications such as drone aerial imaging, remote sensing analysis, intelligent security surveillance, and autonomous driving (Sun et al., 2020; Geiger et al., 2013). However, achieving reliable performance in these domains remains a significant challenge (Li et al., 2021; Yu et al., 2020), as a high rate of false negatives severely degrades recall. This occurs because small targets are essentially lost in a sea of background pixels. Their visual footprint is so minimal that they often appear as little more than textural anomalies or noise within the larger scene. Without strong distinguishing features to separate them, they are easily overwhelmed and absorbed by the surrounding visual context, leading to their frequent misclassification as background. This issue is exacerbated by the design of detectors such as the YOLO series (Redmon & Farhadi, 2018; Bochkovskiy et al., 2020; Ge et al., 2021). In these models, the final confidence score $s(C_i|I)$ for a potential detection of class $C_i$ is typically formulated as the product of the foreground score $p(F|I)$ and the class score $p(C_i|F)$:

$$s(C_i|I) \propto p(F|I) \cdot p(C_i|F), \tag{1}$$

where the foreground score is the probability that a given location corresponds to a foreground objec $F$ conditioned on the image $I$, while the class score is the conditional probability that said object belongs to the $i$-th class, $C_i$. Eq. 1 reveals the core vulnerability. The estimation of the foreground score $p(F|I)$ relies on a local grid's supervision. But a single grid's receptive field is often much larger than the small object it represents, its features are inevitably dominated by the surrounding background context. This inherent ambiguity forces the network to produce suppressed and unreliable foreground scores at inference, as it cannot confidently distinguish the target's faint signal from the dominant background noise. Fig. 1(a) displays an example from the VisDrone dataset (Zhu et al., 2021), showing regions where the foreground scores produced by YOLOX (Ge et al., 2021) are severely suppressed. Even in densely clustered object areas, these scores remain notably subdued, providing strong evidence that conventional detectors indeed encounter significant foreground discrimination errors in high-resolution small object detection tasks.

Early attempts to address the unreliable estimation of the foreground score $p(F|I)$ often relied on a straightforward crop-then-detect strategy (Ozge Unel et al., 2019), as shown in Fig. 1(c). They partitions high-resolution images $I$ into smaller, overlapping patches $\{I_p\}$. By making the object

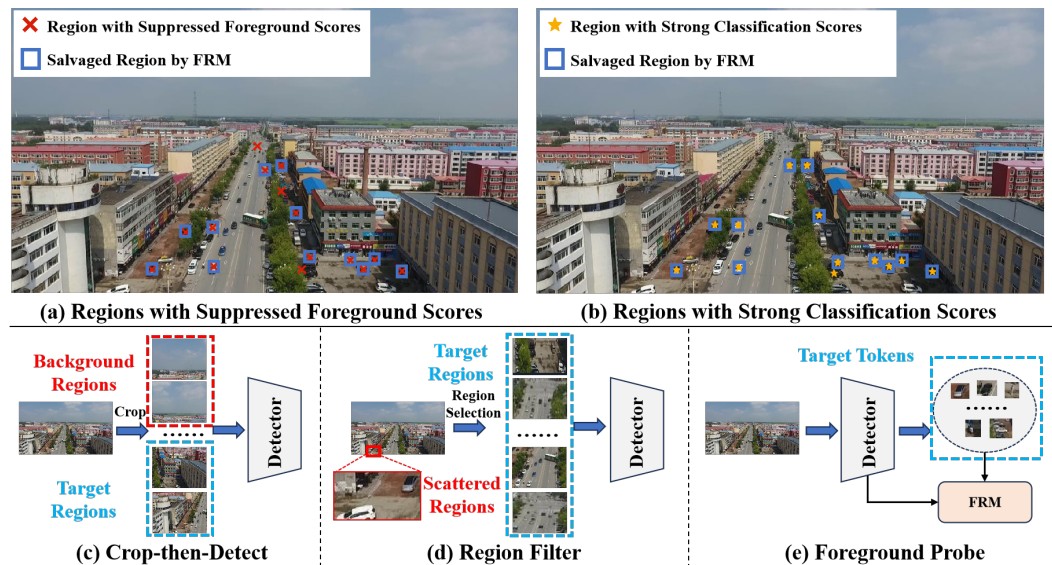

Figure 1: Motivation and comparison of detection paradigms. (a) Standard detectors miss many objects (red crosses) due to weak foreground scores. (b) We observe that classification scores remain strong in these same regions (yellow stars) and our FRM salvages these detections (blue squares). (c-e) A comparison shows our Foreground Probe paradigm (e) is more targeted than the exhaustive Crop-then-Detect (c) or the Region Filter (d) approaches.

relatively larger within a given patch $I_p$, the goal was to make the local foreground score $p(F|I_p)$ more robust and easier to estimate. However, this approach failed to fundamentally address the issue of small targets being overwhelmed by the background, as it merely shifted the problem from the pixel level to the patch level. While this strategy could potentially boost the score for a target within a specific patch, the detector was still forced to process a multitude of patches containing only uninformative background. This meant the core challenge of distinguishing the rare, target-containing patches from the vast majority of empty ones persisted, effectively leaving the problem of low prior object probability unresolved. More critically, objects severed by patch boundaries lost crucial contextual information, hindering the reliable estimation of the class score $p(C_i|F)$ due to incomplete structural and contextual cues.

To mitigate these issues, recent research (Yang et al., 2019; Law et al., 2019; Liu et al., 2024; Koyun et al., 2022) has largely adopted a regions-filter paradigm, as shown in Fig. 1(d). This strategy aims to reduce computational burden by identifying and filtering out irrelevant background regions, allowing detectors to concentrate resources on high-probability target areas, thereby striving for an optimal balance between speed and precision. However, the direct impact of this paradigm on detection accuracy has often been neglected. This is because the paradigm introduces its own set of challenges. Firstly, its performance is heavily contingent upon the quality of the auxiliary region proposal algorithm, making the final detection accuracy dependent on the proposal method's precision. Secondly, the problem of background dominance often persists even within these correctly identified regions. As illustrated by the target regions shown in Fig. 1(d), the small targets can still be so sparse within the crop that their features remain susceptible to being overwhelmed by the immediate background. Additionally, the region proposal mechanism is inherently biased towards dense clusters and is thus prone to ignoring isolated objects. As visually demonstrated in Fig. 1(d), a lone vehicle, explicitly labeled as a "Scattered Region", is excluded from the selection of "Target Regions" by the proposal mechanism, leading to an inevitable missed detection. Thus, the fundamental issues of background information redundancy and foreground-background entanglement remain largely unaddressed.

How can we accurately probe for small objects and enhance their foreground scores? Our solution stems from a key observation that while the foreground confidence in target regions is often severely suppressed, the underlying classification information is not entirely lost and remains comparatively robust. As shown in Fig. 1(a) and (b), a baseline detector produces numerous missed detections (marked with red crosses) where the foreground scores are severely suppressed, rendering the ob-

jects indistinguishable from the background. In stark contrast, an analysis of the classification head reveals that these exact same regions exhibit strong, discriminative scores (marked with yellow stars), indicating that the crucial semantic information is still well-preserved.

Inspired by this, we introduce a novel *Foreground Probing paradigm*. As illustrated in Fig. 1(e), this paradigm moves beyond isolated local estimation by first selecting a sparse set of candidate "Target Tokens" and then refining their foreground scores based on their collective classification features. Specifically, we design a *Sparse Token Selection Module (STSM)* to identify and salvage promising foreground tokens from the sea of background features (*e.g.*, sky, road, buildings), thereby preventing them from being overwhelmed. For instance, an aerial image containing a small cluster of vehicles. A dense detector would place tokens on the vehicles, but also on expanses of empty pavement, rooftops, and vegetation. The STSM's role is to perform a coarse but critical pruning, discarding the tokens on the definite background and retaining only those with a plausible, even if weak, initial signal. This ensures that the FRM's attention mechanism computes semantic affinities among the a set of potential vehicle candidates, rather than having its analysis swamped by irrelevant background features. To further refine the foreground scores, we introduce a *Foreground Refinement Module (FRM)*. The FRM is designed to incorporate more explicit semantic guidance. It distills a semantically rich attention map from the classification features, which then directs the information aggregation and update process for the foreground scores. This allows each token to reference surrounding tokens with similar semantic characteristics. For instance, a candidate token with a weak initial foreground score, if surrounded by other tokens with high semantic similarity, is likely a true positive that has been suppressed. As demonstrated in Fig. 1(a) and (b), our FRM leverages the robust semantic context to correct suppressed foreground scores, successfully salvaging numerous detections that would otherwise be missed by the baseline detector (marked by blue squares).

Our primary contributions can be summarized as follows.

- We introduce the Foreground Probing paradigm, a new framework for small object detection. It fundamentally shifts the focus from unreliable, isolated local predictions to a robust, context-aware refinement process that leverages collective semantic information.

- We propose a novel Sparse Token Selection Module (STSM) that identifies and salvages promising foreground candidates from the sea of background features, preventing them from being prematurely dismissed due to low initial confidence.

- We design a Foreground Refinement Module (FRM) that further refines foreground scores. This module distills a semantic attention map from robust classification features to guide the enhancement of suppressed foreground scores, significantly improving recall.

- Extensive experiments and ablation studies demonstrate the effectiveness of our proposed framework and its superiority over state-of-the-art methods in high-resolution small object detection, particularly in challenging drone-based aerial imagery.

## 2 RELATED WORK

*Vanilla Object Detection.* With the advent of deep neural networks and the availability of large-scale datasets (e.g., COCO (Lin et al., 2014) and PASCAL VOC (Hoiem et al., 2009)), the performance of deep learning-based methods has significantly improved. Traditional object detectors can be broadly categorized into two types: one-stage and two-stage. Two-stage detectors first propose candidate regions and then utilize Region of Interest (ROI) mechanisms to extract features from these regions, subsequently performing bounding box regression and classification prediction. Representative two-stage detectors include the R-CNN (Girshick et al., 2014) series, such as Fast R-CNN (Girshick, 2015) and Faster R-CNN (Ren et al., 2016). While these detectors achieve impressive object detection performance, they often incur a significant computational time burden. Conversely, one-stage detectors employ a single forward pass to directly output prediction results, offering reduced inference time. Prominent examples of one-stage detectors are the YOLO series (Redmon & Farhadi, 2018; Bochkovskiy et al., 2020; Wang et al., 2024b;a) and RetinaNet (Lin et al., 2017). However, a common drawback of one-stage detectors is their reliance on handcrafted components, such as anchor sizes and Non-Maximum Suppression (NMS), which can hinder their generalization ability across different datasets. Following the introduction of the Transformer architecture for Natural Language Processing (NLP) by (Vaswani et al., 2017), Transformer-based architectures have been widely adopted in computer vision. Specifically, (Carion et al., 2020) pioneered the approach of

framing object detection as a set prediction problem and employed Transformers for end-to-end object detection. While DETR-based architectures have revolutionized object detection with their transformer-centric design, they inherently face several limitations. A primary concern is the substantial computational burden imposed by the $O(n^2)$ complexity of the self-attention mechanism, which hinders its application on high-resolution image object detections. Additionally, their one-to-one label assignment strategy often contributes to a relatively slower convergence rate during the training phase. To mitigate these issues, considerable research efforts have been dedicated to enhancing DETR's efficiency and training dynamics. Deformable-DETR (Zhu et al., 2020), for example, proposed a deformable attention mechanism to reduce the computational overhead, particularly within the decoder. Simultaneously, studies by (Chen et al., 2023; Jia et al., 2023; Zong et al., 2023) have explored one-to-many label assignment strategies to achieve faster convergence.

Despite these advancements, accurately detecting small objects in high-resolution images remains a key challenge for vanilla detectors due to two primary obstacles: the prohibitive computational cost of processing large inputs, and the severe scarcity of positive samples assigned during training.

*Small Object Detection*. With the continuous advancement of hardware, several high-resolution object detection datasets have emerged, such as VisDrone (Zhu et al., 2021), UAVDT (Du et al., 2018) and TinyPerson (Yu et al., 2020). These datasets are often characterized by objects that are typically sparse and small, posing significant challenges for conventional detectors. To address these issues, several approaches have been explored. Firstly, data augmentation has proven to be an effective strategy to enhance the performance of standard detectors. For instance, the simple copy-paste method (Ghiasi et al., 2021) specifically aims to mitigate the challenges presented by the sparse and localized nature of tiny objects in high-resolution images. Secondly, the "filter-then-detect" paradigm offers an alternative solution. This approach initially locates plausible regions within images and subsequently performs object detection exclusively within these identified regions. Numerous studies have focused on developing more accurate methods for region localization. For example, ClusDet (Yang et al., 2019) employs a dedicated subnet to predict candidate regions for objects. Similarly, CDMNet (Duan et al., 2021) combines a density map and a segmentation map to generate refined region proposals. ESOD (Liu et al., 2024) generate pseudo labels by SAM (Kirillov et al., 2023) and GT boxes and then select candidate region to avoid useless computational cost caused by background region. While this paradigm significantly reduces computational costs, it often necessitates an additional network for region prediction, and a notable limitation is the lack of interaction or information exchange between different predicted regions. Additionally, (Sun et al., 2025) proposes bounded and confidence-driven gradients to mitigating instability during training and minimize prediction uncertainty. (Xiao et al., 2025; Gong et al., 2021; Fu et al., 2018) also have focused on designing effective feature aggregation modules to enhance feature representation for small object detection.

To address these challenges, this paper introduces a novel Foreground Probing paradigm that directly enhances the detector's foreground discrimination ability by leveraging contextual information.

## 3 METHOD

As illustrated in Fig. 2(a), our Foreground Probing paradigm offers a novel context-aware framework designed to actively identify and refine foreground scores for small objects. Our method is built upon YOLOX (Ge et al., 2021), a strong and widely recognized baseline detector that features a decoupled design for classification and regression features. Given an input image, it first passes through a YOLOX backbone to extract multi-scale feature maps. These feature maps are then fed into the decoupled detection head, which typically consists of separate branches for classification features and regression features. Firstly, the Sparse Token Selection Module (STSM), shown in Fig. 2(b), identifies a sparse set of high-potential foreground candidates across the feature maps. Secondly, as shown in Fig. 2(c), we introduce the Foreground Refinement Module (FRM) that refines the foreground scores of these selected candidates. The FRM leverages the rich contextual relationships embedded within the collective classification features to guide the aggregation of information, producing highly accurate foreground scores. The refined foreground scores, combined with the original regression predictions, then yield the final detection outputs. This approach allows our framework to effectively mitigate the suppression of foreground scores for small objects, thereby significantly improving recall and overall performance in high-resolution small object detection.

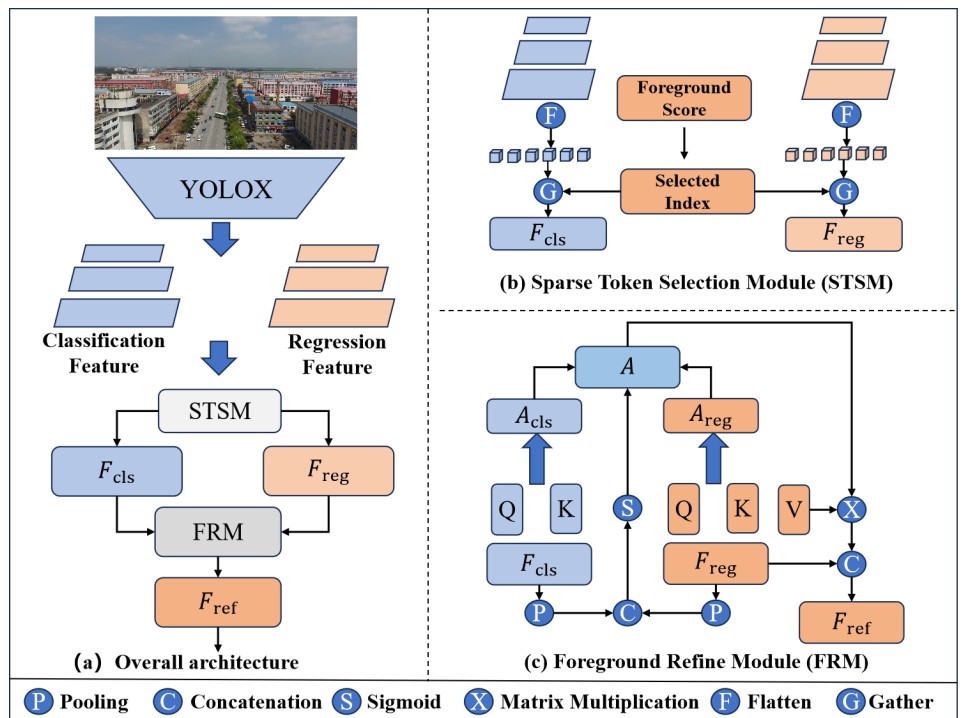

Figure 2: The architecture of our proposed Foreground Probing paradigm. (a) An overview of the pipeline. (b) Sparse Token Selection Module (STSM). (c) Foreground Refinement Module (FRM).

## 3.1 SPARSE TOKEN SELECTION MODULE

Our Sparse Token Selection Module (STSM), shown in Fig. 2(b), is designed to salvage the most promising foreground candidates from this sea of background noise.

STSM takes the direct, pre-NMS (Non-Maximum Suppression) outputs from the YOLOX head. The outputs consists of predictions across $l$ different scale feature maps. This results in a total of $N = \sum_{i=1}^{l} h_i \times w_i$ tokens, where $h_i$ and $w_i$ represent the height and width of the $i$-th feature map, respectively. For each of these $N$ tokens, the raw outputs provide three components: the foreground confidence score $P_{\text{obj}} \in \mathbb{R}^{N \times 1}$, the bounding box regression results $B_{\text{reg}} \in \mathbb{R}^{N \times 4}$ and the classification scores $P_{\text{cls}} \in \mathbb{R}^{N \times C}$, where $C$ is the number of classes.

The STSM leverages the foreground scores $P_{\text{obj}}$ by identifying the set of indices corresponding to the $K$ tokens with the highest foreground scores. Let $P_{\text{obj}} = \{p_1, p_2, \ldots, p_N\}$ be the vector of scores. We define a permutation $\pi$ of the indices $\{1, \ldots, N\}$ that sorts these scores in descending order, such that $p_{\pi(1)} \geq p_{\pi(2)} \geq \cdots \geq p_{\pi(N)}$. The set of selected indices, $\mathcal{I}_s$, is then formally defined as the first $K$ elements from this sorted permutation:

$$\mathcal{I}_s = \{\pi(i) \mid i = 1, \ldots, K\}. \tag{2}$$

This process effectively identifies a sparse set of candidate tokens that are most likely to contain objects, even if their initial confidence is low. The value of $K$ is a hyperparameter. A larger $K$ increases the probability of including all true positives but also incorporates more background noise for the FRM to process. Through empirical validation, we found $K = 500$ to provide an optimal balance, ensuring high recall while significantly reducing the computational burden.

Once the candidate indices $\mathcal{I}_s$ are identified, we use them to gather the corresponding features from the complete set of predictions. This results in the creation of a sparse set of classification features $F_{\text{cls}} = \{f_{\text{cls}}^i \mid i \in \mathcal{I}_s\}$ and a corresponding set of regression features $F_{\text{reg}} = \{f_{\text{reg}}^i \mid i \in \mathcal{I}_s\}$. These curated sets of features are then passed to the Foreground Refinement Module for more sophisticated, context-aware processing. By converting the dense prediction problem into a sparse refinement task, the STSM serves as the critical first step in our paradigm, efficiently identifying and isolating promising candidates for focused enhancement.

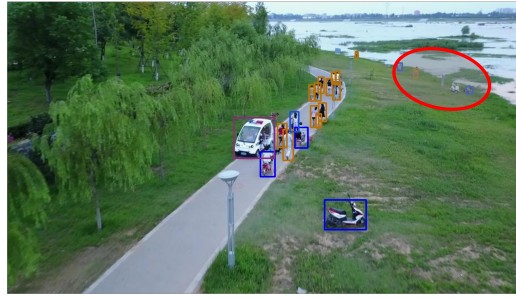

**(a) w/o Foreground Refinement Module**    **(b) w/ Foreground Refinement Module**

Figure 3: Effectiveness of our Foreground Refinement Module (FRM) in a challenging scenario. Compared to the baseline YOLOX (a) (Ge et al., 2021), our FRM-enhanced model (b) successfully recovers numerous missed detections within the highlighted red region.

### 3.2 FOREGROUND REFINEMENT MODULE

Following the sparse selection of $K$ promising tokens by the STSM, the Foreground Refinement Module (FRM) receives the corresponding sets of features, as shown in Fig. 2(c). Let $F_{\text{reg}} \in \mathbb{R}^{K \times D_{\text{reg}}}$ and $F_{\text{cls}} \in \mathbb{R}^{K \times D_{\text{cls}}}$ denote the feature vectors used by the detector's head to predict foreground scores and classification scores, respectively. As established in our motivation, the foreground features in $F_{\text{obj}}$ are often ambiguous and lead to suppressed scores, while the classification features in $F_{\text{cls}}$ retain more robust semantic information. The objective of the FRM is to leverage the rich context within $F_{\text{cls}}$ to enhance and re-evaluate the features in $F_{\text{obj}}$. This fundamental disparity between suppressed foreground scores and robust classification scores, which forms the empirical basis for our FRM, is visually demonstrated in Fig. 1(a) and (b).

Firstly, we project the classification features $F_{\text{cls}}$ into query ($Q_{\text{cls}}$) and key ($K_{\text{cls}}$) spaces, and the foreground features $F_{\text{reg}}$ into query ($Q_{\text{reg}}$), key ($K_{\text{reg}}$) and value ($V_{\text{reg}}$) spaces, respectively. Next, we compute the self-attention scores for both regression and classification features independently, following the standard scaled dot-product attention mechanism (Vaswani et al., 2017). The attention scores quantify the pairwise relationships between the $K$ features within each set:

$$A_{\text{reg}} = \text{Softmax}(Q_{\text{reg}} K_{\text{reg}}^T / \sqrt{d_k}), \quad A_{\text{cls}} = \text{Softmax}(Q_{\text{cls}} K_{\text{cls}}^T) / \sqrt{d_k}, \quad (3)$$

$A_{\text{reg}} \in \mathbb{R}^{K \times K}$ and $A_{\text{cls}} \in \mathbb{R}^{K \times K}$ capture the internal dependencies and relationships among the $K$ features within the regression and classification feature sets, respectively. $A_{\text{reg}}$ indicates how each regression feature relates to every other regression feature, while $A_{\text{cls}}$ highlights the salient relationships among features from the perspective of the classification task.

The core of FRM is to adaptively combine these two attention maps. We introduce a learnable gating mechanism to control the influence of classification attention on the final refined attention. A gated score $\lambda$ is obtained by concatenating the global representations of $F_{\text{reg}}$ and $F_{\text{cls}}$, followed by a linear layer and a sigmoid activation. Since $F_{\text{reg}}$ and $F_{\text{cls}}$ are sets of features, we first aggregate them into single vectors $f_{\text{reg}}$ and $f_{\text{cls}}$ using a global average pooling operation across the $K$ features:

$$f_{\text{reg}} = \text{AvgPool}(F_{\text{reg}}), \quad f_{\text{cls}} = \text{AvgPool}(F_{\text{reg}}), \quad \tilde{\lambda} = \text{Concat}(f_{\text{reg}}, f_{\text{cls}}), \quad \lambda = \sigma(\tilde{\lambda} W_\lambda), \quad (4)$$

where $\text{Concat}(\cdot, \cdot)$ represents the concatenation of channel dimensions. $\sigma$ denotes the sigmoid activation, ensuring that $\lambda$ is a scalar value between 0 and 1, acting as a gating score. Specifically, a higher $\lambda$ indicates a stronger reliance on the regression's self-attention, while a lower $\lambda$ suggests that the classification's attention provides more valuable guidance. The final combined attention matrix $A \in \mathbb{R}^{K \times K}$ is calculated as a weighted sum of $A_{\text{reg}}$ and $A_{cls}$:

$$A = \lambda \cdot A_{\text{reg}} + (1 - \lambda) \cdot A_{\text{cls}}. \quad (5)$$

This adaptive combination allows the module to dynamically adjust the contribution of classification-derived attention based on the current feature states, ensuring that the regression features benefit from the classification's discriminative focus without being entirely dominated by it. For instance, if the classification task strongly highlights the relationship between specific features

Table 1: Performance comparison with state-of-the-art methods on the UAVDT (Du et al., 2018) dataset. The best results are in **bold**, and the second-best are underlined.

| Detector | Backbone | AP | AP$_{50}$ | FPS |
|---|---|---|---|---|
| ClusDET (Yang et al., 2019) | ResNet50 | 13.7 | 26.5 | 7.2 |
| DMNet (Li et al., 2020) | ResNet50 | 14.7 | 24.6 | 6.7 |
| CDMNet (Duan et al., 2021) | ResNet50 | 20.7 | 35.5 | - |
| CEASC (Du et al., 2023) | ResNet18 | 17.1 | 30.9 | 29.4 |
| ESOD (Liu et al., 2024) | YOLOV5 | 22.5 | 40.7 | 32.8 |
| FBRT-YOLO (Xiao et al., 2025) | YOLOV8 | 18.4 | 31.1 | - |
| **Ours** | CSPDarkNet | **23.9** | **41.2** | 16.4 |

(*e.g.*, features corresponding to a particular object part), this combined attention will guide the regression task to also focus on those inter-feature relationships.

Finally, the refined regression feature set $\tilde{F}_{\text{reg}}$ is obtained by applying this combined attention to the regression value features $V_{\text{reg}}$, and then concatenating the result with the original regression features $F_{\text{reg}}$. This concatenation allows the refined features to retain their original information while incorporating the attention-weighted contextual information.

$$\tilde{F}_{\text{reg}} = \text{Concat}(F_{\text{reg}}, A \cdot V_{\text{reg}}). \tag{6}$$

The output $\tilde{F}_{\text{reg}} \in \mathbb{R}^{N \times (D_{\text{reg}} + d_v)}$ then serves as the input for subsequent regression prediction layers. By integrating the classification-guided attention, the FRM effectively encourages the regression branch to attend to individual features and their relationships that are not only relevant for its own task but also discriminative and salient from the perspective of the classification task, thereby leading to more robust and accurate regression predictions. The practical efficacy of this refinement is starkly illustrated in Fig. 3, where our FRM enables the model to salvage numerous detections in a challenging scene that are entirely missed by the baseline detector.

### 3.3 LOSS FUNCTION

After the FRM produces the refined foreground features, a final prediction head generates the enhanced foreground scores. During training, the entire network is optimized using a composite loss function $\mathcal{L}$, defined as a weighted sum of four components:

$$\mathcal{L} = \lambda_{\text{ref}}\mathcal{L}_{\text{ref}} + \lambda_{iou}\mathcal{L}_{\text{iou}} + \lambda_{\text{cls}}\mathcal{L}_{\text{cls}} + \lambda_{\text{obj}}\mathcal{L}_{\text{obj}}, \tag{7}$$

where $\lambda_{\text{ref}}$, $\lambda_{\text{iou}}$, $\lambda_{\text{cls}}$, and $\lambda_{\text{obj}}$ are coefficients that balance the contribution of each task.

The $\mathcal{L}_{\text{ref}}$, is specifically designed to train our Foreground Refinement Module. This loss is computed exclusively on the $K$ candidate tokens selected by the STSM. For each of these $K$ tokens, a ground-truth objectness label $y_{\text{ref}} \in \{0, 1\}$ is assigned. The loss is then formulated as a Binary Cross-Entropy loss between these labels and the refined foreground scores $\hat{P}_{\text{ref}}$ from the FRM:

$$\mathcal{L}_{\text{ref}} = -\frac{1}{K} \sum_{k=1}^{K} \left[ y_{\text{ref}}^{(k)} \log(\hat{P}_{\text{ref}}^{(k)}) + (1 - y_{\text{ref}}^{(k)}) \log(1 - \hat{P}_{\text{ref}}^{(k)}) \right]. \tag{8}$$

This focused supervision effectively teaches the FRM to leverage context to boost the scores of true positives and suppress the scores of hard negatives within the high-potential candidate set.

The other components, $\mathcal{L}_{\text{iou}}$, $\mathcal{L}_{\text{cls}}$ and $\mathcal{L}_{\text{obj}}$, are standard losses that supervise the baseline detector's predictions on positive samples identified by the label assignment strategy. For localization, $\mathcal{L}_{\text{iou}} = \mathbb{E}_{i \in \text{Pos}} [1 - \text{CIoU}_i]$, where $\mathbb{E}_{i \in \text{Pos}}$ denotes the mean over the set of positive samples Pos, and $\text{CIoU}_i$ is the Complete Intersection over Union for the i-th sample. The classification loss $\mathcal{L}_{\text{cls}}$ and the initial objectness loss $\mathcal{L}_{\text{obj}}$ are both based on Binary Cross-Entropy (BCE). $\mathcal{L}_{\text{cls}}$ is applied to the multi-class predictions to penalize incorrect class probabilities, while $\mathcal{L}_{\text{obj}}$ supervises the base detector's initial foreground probability scores for the same set of positive samples.

Table 2: Performance comparison with state-of-the-art methods on the VisDrone (Zhu et al., 2021) dataset. The best results are in **bold**, and the second-best are underlined.

| Detector | Backbone | AP | AP$_{50}$ | FPS |
|---|---|---|---|---|
| ClusDET (Yang et al., 2019) | ResNeXt101 | 32.4 | 56.2 | 6.3 |
| DMNet (Li et al., 2020) | ResNet101 | 28.5 | 48.1 | 5.9 |
| CDMNet (Duan et al., 2021) | ResNeXt101 | 31.9 | 52.9 | - |
| UFPMP-Det (Huang et al., 2022) | ResNet50 | 36.6 | 62.4 | 8.5 |
| QueryDet (Yang et al., 2022) | ResNet50 | 28.3 | 48.1 | 8.6 |
| CEASC (Du et al., 2023) | ResNet18 | 28.7 | 50.7 | 26.9 |
| ESOD (Liu et al., 2024) | YOLOV5 | 36.0 | 59.7 | 36.4 |
| UGS (Sun et al., 2025) | ResNet50 | 38.1 | 61.9 | - |
| YOLOX | CSPDarkNet | 36.8 | 59.3 | 16.2 |
| **Ours** | CSPDarkNet | **39.1** | **62.7** | 14.5 |

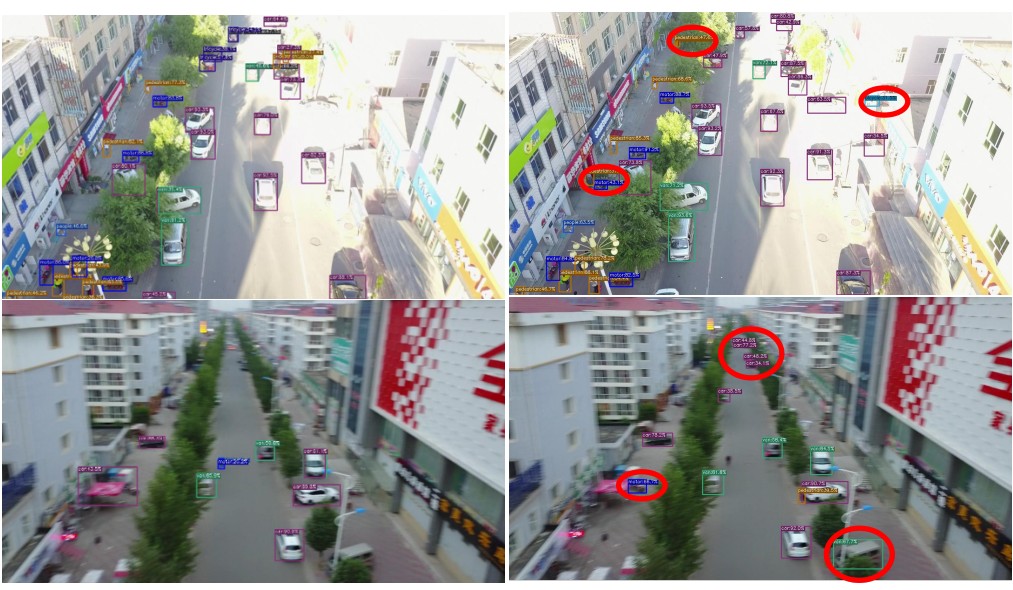

(a) ESOD (Liu et al., 2024)            (b) Ours

Figure 4: Qualitative comparison with the state-of-the-art method ESOD (Liu et al., 2024).

## 4 EXPERIMENTS

### 4.1 EXPERIMENTAL SETTINGS

We implement our method based on YOLOX-X (Ge et al., 2021) with 4 scales detection head on two NVIDIA RTX3090 GPUs with a batchsize of 8. We set initial learning rate as 0.01 for 100 epochs. The input size of image in training is set (960, 960) and test image sizes of VisDrone and UAVDT are set as (1504, 1504) and (1286, 1286) following the setting of ESOD (Liu et al., 2024). We conduct the experiments on two different datasets: VisDrone (Zhu et al., 2021) and UAVDT (Du et al., 2018). More detailed description can be found in Appendix A.1.

### 4.2 COMPARISON WITH STATE-OF-THE-ART METHODS

*Quantitative Comparison..* We benchmark our method against state-of-the-art (SOTA) detectors on the VisDrone(Zhu et al., 2021) dataset, with the results detailed in Table 2. Our approach demonstrates superior accuracy with only a marginal increase in computational overhead. Specifically, when compared to UGS (Sun et al., 2025), which is built upon a 5-scale DINO (Zhang et al., 2022) backbone, our method achieves a notable 1.0% improvement in AP. Furthermore, our method significantly outperforms detectors that rely on heavier backbones like ResNeXt-101 or ResNet-101, surpassing ClusDet (Yang et al., 2019) by 6.7% AP and CDMNet (Duan et al., 2021) by 7.9% AP.

Table 3: Effectiveness of selected token number

| N | 100 | 300 | 500 | 700 |
|---|---|---|---|---|
| AP | 37.7 | 38.4 | 39.1 | 39.1 |
| AP50 | 60.3 | 61.7 | 62.7 | 62.7 |

Table 4: Effectiveness of selected Gated Score

| $\lambda$ | 0.3 | 0.5 | 0.7 | learnable |
|---|---|---|---|---|
| AP | 37.6 | 38.5 | 38.0 | 39.1 |
| AP50 | 60.4 | 61.9 | 61.0 | 62.7 |

While UFPMP-Det surpasses our method by 2.4% on AP50, this advantage comes at the cost of a severe computational burden, as it employs a complex dual-detector architecture. Additionally, our module enhances its YOLOX baseline by a significant 3.1% AP with a minimal speed reduction of only 1.7 FPS.

To demonstrate the generalizability of our approach, we also evaluated it on the UAVDT (Du et al., 2018) dataset, with results presented in Table 1. Here, our method continues to show significant improvements over SOTA. It surpasses ESOD (Liu et al., 2024) by 1.4% AP and FBRT-YOLO (Xiao et al., 2025) by 3.7% AP. These consistent gains across two distinct datasets underscore the robustness and effectiveness of our proposed module.

*Qualitative comparison.* To complement the quantitative results, we provide a qualitative comparison between our method and ESOD (Liu et al., 2024) in Fig. 4. The figure displays ESOD's (Liu et al., 2024) detection results in the left column and ours in the right, with red circles highlighting objects that our method successfully detects but ESOD (Liu et al., 2024) misses. The first row demonstrates a challenging scenario where a bicycle is camouflaged by over-exposure and several pedestrians are obscured in the shade. While ESOD (Liu et al., 2024) fails to detect these objects, our method accurately localizes them. The second row presents a case affected by severe motion blur, which significantly complicates foreground discrimination. Despite this degradation, our method demonstrates its robustness by successfully identifying distant vehicles that ESOD (Liu et al., 2024) overlooks. These examples underscore the effect of our proposed Foreground Refine Module. More cases can be found in Appendix A.4.

### 4.3 ABLATION STUDIES

*Effect of the Number of Selected Tokens.* To analyze the effect of the number of selected tokens, we conducted an ablation study varying this parameter from 100 to 700. The results, presented in Table 3, show that performance on the VisDrone (Zhu et al., 2021) dataset improves substantially as the number of tokens increases up to a certain threshold. Specifically, increasing the token count from 100 to 500 boosts the AP from 37.7 to 39.1 and the AP50 from 60.3 to 62.7. However, a further increase from 500 to 700 tokens yields diminishing returns, with no significant improvement in detection accuracy.

*Effect of Gated Score $\lambda$.* Our proposed FRM incorporates a learnable parameter, $\lambda$, which we refer to as the Gated Score. This parameter adaptively controls the degree of influence from the classification feature attention map. To validate the effectiveness of making $\lambda$ learnable, we conducted an ablation study comparing it against several fixed values, with results shown in Table 4. With a fixed parameter, the peak performance of 38.5% AP was achieved at $\lambda = 0.5$. Deviating from this value, either lower or higher, resulted in degraded performance. Crucially, all fixed-value settings were outperformed by our proposed learnable approach, which demonstrates the clear benefit of allowing the model to dynamically tailor the influence of the attention map.

## 5 CONCLUSION

In this work, we address a critical failure in small object detection: the suppression of foreground scores that leads to missed targets. Our solution is built upon the key insight that underlying classification features remain robust even when foreground signals are weak. Inspired by this, we introduce the novel Foreground Probing paradigm, which employs a Sparse Token Selection Module (STSM) to salvage promising candidates from background noise and a Foreground Refinement Module (FRM) to refine their scores using collective semantic context. Extensive experiments on challenging benchmarks demonstrate that our method establishes a new state-of-the-art in both accuracy and efficiency. This proves that by shifting the focus from isolated estimation to context-aware refinement, our approach provides a powerful and effective strategy for unmasking small objects.

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

# A    APPENDIX

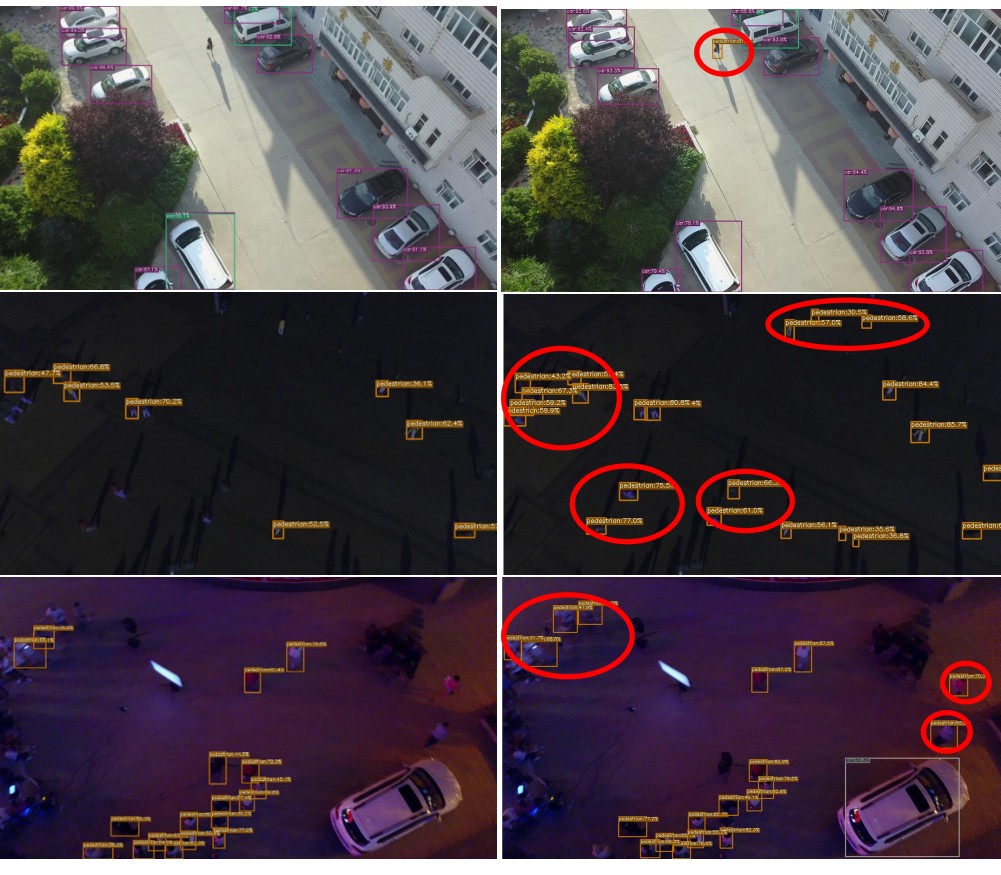

(a) ESOD (Liu et al., 2024)          (b) Ours

Figure 5: More comparison with the state-of-the-art method ESOD (Liu et al., 2024).

## A.1    DATASETS

*VisDrone* (Zhu et al., 2021), a comprehensive collection of 10209 images, predominantly at 1920x1080 pixels, captured from various drone platforms. The visual content within these images is rich and varied, showcasing bustling city streets, open roads, and residential areas. Each image is meticulously annotated with bounding boxes for 10 distinct object categories. Furthermore, the images consistently present densely clustered objects, where individual small instances are closely packed, alongside instances of severe occlusions and a wide range of lighting and weather conditions. This unique combination of high resolution and pervasive small, challenging targets makes VisDrone an indispensable benchmark for developing and evaluating robust small object detection algorithms for aerial images.

*UAVDT* (Du et al., 2018), large-scale benchmark for object detection and tracking in videos captured by Unmanned Aerial Vehicles (UAVs). It comprises 50 video sequences with a resolution of 1024×540, which are split into 30 for training and 20 for testing. The dataset focuses on three vehicle categories (car, truck, and bus), with an average of 18 annotated objects per frame. A key feature of UAVDT is its rich set of attributes, including weather conditions (daylight, night, fog), flying altitude, camera view, and occlusion levels. This detailed annotation facilitates a comprehensive evaluation of algorithm performance and robustness under diverse conditions.

## A.2    LIMITATIONS

While our method demonstrably enhances the detector's foreground probing capabilities, we acknowledge that certain challenging cases remain unresolved. As illustrated in Fig 6, our model ex-

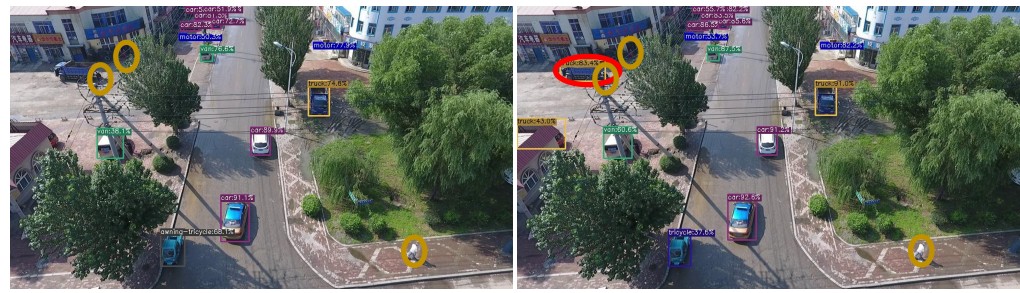

(a) ESOD (Liu et al., 2024)                    (b) Ours

Figure 6: Limitation of our method

hibits limitations in complex scenes with multiple failure modes. For instance, although our method successfully identifies a truck previously missed by ESOD (Liu et al., 2024) (highlighted by the red circle), it fails to detect three other foreground objects in the same image.

Two people on the left are overlooked, likely due to their extremely small scale, which causes their features to become indistinguishable from the surrounding background textures. Furthermore, a pedestrian in the bottom-right corner is misclassified as background because their appearance exhibits extremely low contrast with the floor. These failure cases underscore a fundamental challenge: robustly perceiving objects under extreme variations in scale and appearance. Developing a more advanced mechanism to address this will be the primary focus of our future work.

## A.3 MORE VISUAL COMPARISON

Further visual comparisons are dispayed in Fig 5, which highlight our method's advantages over ESOD, with newly detected targets circled for clarity. For instance, our approach successfully identifies a pedestrian that ESOD (Liu et al., 2024) misclassifies as background (first row). It also excels in challenging low-light conditions, accurately detecting pedestrians obscured by darkness where ESOD (Liu et al., 2024) fails (second row). In each of these difficult cases, our proposed module proves its superior accuracy and robustness.

## A.4 ACKNOWLEDGEMENT

We would like to thank Gemini, a large language model from Google, for its valuable assistance in the language polishing and refinement of this manuscript. All core ideas, experimental work, and conclusions presented in this paper are the original contributions of human authors.

