# OpenReview forum: "Unmasking the Tiny: Foreground Probing for Small Object Detection"
_ICLR.cc/2026/Conference — ICLR 2026 Conference Withdrawn Submission_

### Official Review · Reviewer_gREz · 2025-10-25

**Soundness:** 3
**Presentation:** 3
**Contribution:** 2
**Rating:** 4
**Confidence:** 5

**Summary:**

This paper proposes a new paradigm for Small Object Detection (SOD) called Foreground Probing (FP). Through two core modules—Sparse Token Selection Module (STSM) and Foreground Refinement Module (FRM)—the authors attempt to recover the suppressed foreground confidence from the semantic features of the classification branch, thereby improving the recall rate of small objects. Experiments on the VisDrone and UAVDT datasets demonstrate consistent performance improvements.

**Strengths:**

1. Clear problem definition and motivation: The paper accurately identifies the key bottleneck in current small object detection: the foreground score is suppressed by the large background. The analytical formulation is insightful, theoretically revealing the root cause of confidence suppression in detectors.

2. Simple and highly compatible design: The combination of STSM and FRM is easy to integrate into existing detection frameworks (such as YOLOX) without modifying the backbone architecture, making it suitable for large-scale deployment.

**Weaknesses:**

1. The paper does not provide a visual comparison between classification scores and regression scores, which makes the motivation explanation insufficient.
2. STSM is essentially a top-K selection based on score ranking; it does not introduce a new feature learning mechanism or adaptive selection strategy. It is suggested to explore a learnable token selection strategy for STSM rather than simple ranking, or to include comparative experiments with alternative approaches.
3. The attention weighting mechanism in FRM (A = λA_reg + (1−λ)A_cls) is highly similar to existing cross-attention or gating mechanisms.
4. The paper claims that the semantic features from the classification branch are more stable, but this is not quantitatively verified.

**Questions:**

reference Weakness

---

### Official Review · Reviewer_3s3c · 2025-10-25

**Soundness:** 2
**Presentation:** 2
**Contribution:** 2
**Rating:** 4
**Confidence:** 4

**Summary:**

The paper tackles small-object detection in high-resolution imagery by reframing the failure mode as “suppressed foreground scores” in one-stage, decoupled-head detectors. It introduces a Foreground Probing paradigm built on YOLOX: (i) a Sparse Token Selection Module (STSM) that keeps the top-K candidate tokens by objectness from all feature maps, and (ii) a Foreground Refinement Module (FRM) that uses classification-feature attention to refine the foreground/objectness branch via a gated combination of regression- and classification-self-attentions. Integrated into YOLOX-X, the method reports improvements on VisDrone with a small FPS drop, and improvements on UAVDT over ESO

**Strengths:**

Task-tailored designs for small object detection.

Promising improvement over baseline.

Balanced paper organization.

**Weaknesses:**

There are weaknesses regarding novelty, clarity, and presentation.
1. The method is only designed for single-stage YOLO-style detectors that use separate classification and localization heads. It does not apply to two-stage models like Mask R-CNN or transformer-based models such as DETR and RT-DETR, which already handle small objects quite well through better region proposals or global attention. The paper should clearly state this limitation in the title or abstract. Right now, the writing sometimes gives the impression that the method can improve any type of detector. The authors should also compare or at least discuss more recent DETR-based small-object detectors.

2. Figure 1 does not clearly explain what the “foreground probe” actually is or how it works. The terms “foreground probe” and “foreground refinement” are introduced but not illustrated in an intuitive way. As it stands, the figure is too abstract to help readers understand the mechanism.

3. The experimental comparisons do not fully support the claimed advantages. The competing methods often use weaker or older backbones, making the improvements less meaningful. Also, efficiency comparisons (FPS) are not clearly standardized. FPS varies greatly depending on GPU type, input size, and framework. The results would be more convincing if the authors compared models with similar settings and included standard measures like parameters, FLOPs, and latency on the same hardware. It would also help to report results by object size (small/medium/large, like in COCO, if possible) to show that the proposed method truly improves detection for small targets.

5. All results are based on YOLOX. It is unclear whether the proposed modules can be easily applied to other detectors.

**Questions:**

In addition to the concerns listed in weaknesses, another question is: the gate function in Eq. (4) seems to contain a typo: fcls should come from Fcls rather than Freg (correct me if I am wrong). The paper does not explain whether the gate is applied globally or per token, which affects the interpretation.

---

### Official Review · Reviewer_Ggyz · 2025-10-25

**Soundness:** 3
**Presentation:** 2
**Contribution:** 3
**Rating:** 4
**Confidence:** 4

**Summary:**

This paper addresses the challenge of detecting small objects in high-resolution images and proposes a Foreground Probing paradigm, which aims to recover suppressed foreground confidence through collective semantic features. The method’s effectiveness is validated through experiments on the VisDrone and UAVDT datasets.

**Strengths:**

# **Strengths**

1. **Clear and Reasonable Idea:** The paper analyzes the internal mechanisms of detectors and identifies that the bottleneck for small object detection lies in the "systematic suppression of foreground confidence." It proposes a solution to feed classification features back to improve foreground estimation. This approach is novel and distinct from traditional "filter-and-detect" or "crop-and-detect" paradigms.

2. **Sound Experiments:** The authors conduct comprehensive evaluations on two mainstream high-resolution UAV datasets (**VisDrone** and **UAVDT**) and perform ablation studies on key hyperparameters (e.g., number of tokens, gating parameter λ). The results show robustness to hyperparameter variations and consistent performance improvements.

3. **Good Readability:** The paper is well-structured, with clear motivation, intuitive figures, and straightforward comparative experiments.

4. **Practical Significance:** Small object detection has broad applications in UAV security, remote sensing, and traffic perception. The proposed improvement strategy is practically valuable.

**Weaknesses:**

# **Weaknesses**

1. **Limited Depth of Innovation:** Although the "Foreground Probing" paradigm provides a new perspective, its core implementation (based on token selection and attention fusion) is still a combination of existing mechanisms rather than a fundamentally new theoretical framework. There is a lack of evaluation on the transferability of this paradigm to different detector architectures, such as Transformer-based or anchor-free DETR series.

2. **Incomplete Baseline Comparisons:** Experiments only compare with a few one-stage detectors like YOLOX and ESOD, without including recent detectors specifically designed for tiny object detection or other mainstream detectors.

3. **Insufficient Ablation Studies:** Ablation studies only cover the number of tokens and the gating parameter λ, without further investigation of other parameters or structural components.

4. **Limited Generalization and Scenario Coverage:** Experiments are confined to UAV aerial datasets (**VisDrone** and **UAVDT**), limiting evaluation of the method’s generalization to other scenarios.

5. **Other Formatting Issues:** Table 2 appears before Table 1 and is cited first; Figure 4’s corresponding dataset is not explicitly specified.

**Questions:**

# **Questions**

1. **Computation Efficiency:** For STSM, the K value (500) and the attention computation in FRM (O(K²))—do they become speed bottlenecks? Could Table 3 include FPS results for different K values (100/300/500/700) to quantify the trade-off between token number, speed, and accuracy?

2. **Experimental Extension:** Could the authors compare with more mainstream small object detection methods, such as RT-DETR or Deformable DETR?

3. **Sensitivity of K and Gating Coefficient λ:** Are the K value and gating coefficient λ consistent across different datasets, or do they require dataset-specific tuning?

4. **Validation in Diverse Scenarios:** Only two datasets are used, which may be insufficient to demonstrate generalization, especially as UAVDT has only three categories and high frame similarity. Have the authors considered evaluating on more diverse scenarios, such as **DOTA**, **TinyPerson**, or other small object detection datasets?

5. **Inference Efficiency:** Tables 1 and 2 show that the baseline YOLOX and the proposed method achieve only 16 FPS. It is recommended to consider stronger baselines to improve efficiency.

6. **Incomplete Ablation Studies:** Ablation experiments only cover K values and the gating parameter λ. Further investigation into other parameters or structural components is suggested.

---

### Note · Authors · 2025-12-31

I have read and agree with the venue's withdrawal policy on behalf of myself and my co-authors.